# SCALABLE MONOTONIC NEURAL NETWORKS

**Hyunho Kim, Jong-Seok Lee**[*]
Department of Industrial Engineering
Sungkyunkwan University
Suwon, Republic of Korea
`{retna319,jongseok}@skku.edu`

## ABSTRACT

In this research, we focus on the problem of learning monotonic neural networks, as preserving the monotonicity of a model with respect to a subset of inputs is crucial for practical applications across various domains. Although several methods have recently been proposed to address this problem, they have limitations such as not guaranteeing monotonicity in certain cases, requiring additional inference time, lacking scalability with increasing network size and number of monotonic inputs, and manipulating network weights during training. To overcome these limitations, we introduce a simple but novel architecture of the partially connected network which incorporates a 'scalable monotonic hidden layer' comprising three units: the exponentiated unit, ReLU unit, and confluence unit. This allows for the repetitive integration of the scalable monotonic hidden layers without other structural constraints. Consequently, our method offers ease of implementation and rapid training through the conventional error-backpropagation algorithm. We accordingly term this method as Scalable Monotonic Neural Networks (SMNN). Numerical experiments demonstrated that our method achieved comparable prediction accuracy to the state-of-the-art approaches while effectively addressing the aforementioned weaknesses.

## 1 INTRODUCTION

The field of neural networks has made significant progress in recent years, leading to notable advancements in domains like computer vision (Redmon & Farhadi, 2018), natural language processing (Vaswani et al., 2017), and speech recognition (Baevski et al., 2020). Neural networks have been successfully applied to a wide range of practical applications, including prediction (Chen et al., 2018; Arik & Pfister, 2021), generation (Goodfellow et al., 2020; Ho et al., 2020; Kobyzev et al., 2020), out-of-distribution detection (Liang et al., 2017), and more. However, the incorporation of prior domain knowledge into neural network models remains an open question. One important inductive bias sought in neural network models is monotonicity, ensuring that the output consistently increases or decreases with changes in certain inputs. In domains where model credibility is critical, such as finance, healthcare, legal, and engineering, it is essential to consider monotonicity when designing neural network models. For instance, in the financial domain, accurately reflecting the positive influence of factors on price is crucial. Similarly, in engineering, predictive models must account for monotonic relationships between control factors and responses to ensure their safe use.

In traditional machine learning, various approaches have been employed to enforce monotonicity in trained models, including decision trees (Potharst & Feelders, 2002), random forests (Bartley et al., 2019), boosting (Chen et al., 2015), support vector machines (Doumpos & Zopounidis, 2009), and traditional statistical regression (Dykstra et al., 2012; Kalai & Sastry, 2009; Kyng et al., 2015). Similarly, there have been research efforts in the past and more recently to incorporate monotonicity as an inductive bias into neural networks (Archer & Wang, 1993; Sill, 1997; Lang, 2005; Daniels & Velikova, 2010; Milani Fard et al., 2016; You et al., 2017), which can be categorized into two groups.

**Approaches with customized architecture:** The first group of studies focuses on modifying neural networks to achieve monotonicity or partial monotonicity in specific input features. Hand-designed

---

[*]Corresponding author

architectures or heuristic algorithms are employed for this purpose. For instance, the non-negative approach (Archer & Wang, 1993) restricts weights on monotonic features to be positive. The Min-Max Network (Daniels & Velikova, 2010) utilizes three specially designed layers (linear layer, Max-pooling layer, Min-pooling layer) to enforce monotonicity. The Deep Lattice Network (You et al., 2017) constructs a network using lattice layers and combines them in an ensemble. Additionally, the Hierarchical Lattice Layer (HLL) method (Yanagisawa et al., 2022) extends the existing deep lattice network to address its limitations, particularly in handling high-dimensional inputs. However, HLL still faces the challenge of significantly increased computational complexity as the dimension of the monotonic features grows. It has been recognized that a common drawback of the aforementioned studies lies in the often excessive constraints imposed, which result in a restricted hypothesis space for weight parameters. Consequently, models in this category tend to exhibit relatively lower performance when compared to the other group. However, recent works have demonstrated their ability to overcome this limitation. The Lipschitz Monotonic Networks (LMN) method (Nolte et al., 2022) has exhibited enhanced performance by constraining the norm of weights, ensuring the trained network maintains Lipschitz continuity. Nevertheless, it is important to note that if the Lipschitz constant $\lambda$ remains smaller than the inherent monotonicity scale within data, LMN may struggle to effectively learn from the data. This limitation arises because LMN cannot approximate any function that holds a gradient exceeding the given Lipschitz constant. Another notable study demonstrating promising performance is the Constrained Monotonic Neural Networks (Constrained MNN) method (Runje & Shankaranarayana, 2023). In this work, a monotonic dense layer was introduced to enforce monotonicity by converting updated weights into absolute values during the training process. The main strength of the approaches within this category lies in their capacity to ensure monotonicity. However, limitations exist. Earlier studies exhibited low performance, while recent advancements come with their own constraints, notably the non-optimality of trained networks resulting from weight manipulation during the training process.

**Approaches with regularization:** The second group aims to achieve monotonicity through regularization techniques. One approach is to impose penalties on negative gradients or directly penalize negative weights of the network as a form of regularization loss (Gupta et al., 2019; Liu et al., 2020; Monteiro et al., 2021). The Certified Monotonic Neural Networks (Certified MNN) method (Liu et al., 2020) penalizes negative gradients for each feature in the data. It leverages the piece-wise linear property of ReLU to verify the monotonicity of the trained network using a mixed-integer linear programming (MILP) solver. However, it requires additional processes to enforce and verify monotonicity. As a regularization method, this method does not always ensure monotonicity. Therefore, in scenarios where the guarantee of monotonicity is essential, opting for the approaches with customized architecture may be preferable. Moreover, as the network depth increases, satisfying the monotonicity constraint becomes more challenging, resulting in significantly higher computational costs. Another regularization approach involves data augmentation. The Counterexample-Guided Learning of Monotonic Neural Network (COMET) method (Sivaraman et al., 2020) iteratively injects monotonicity into the model through guided learning with the augmentation of counterexamples for samples violating monotonicity. Monotonicity guarantees are achieved by finding the upper and lower envelopes of the model using iterative algorithms proposed in this research. The COMET method utilizes the satisfiability modulo theories (SMT) solver (Barrett & Tinelli, 2018) to find counterexamples. Similar to Certified MNN, COMET does not employ end-to-end learning, and as the network size grows, the solver becomes computationally intensive. Both Certified MNN and COMET have the advantage of exploring weights within an unrestricted hypothesis space. However, they also share the disadvantage of requiring additional steps in the inference or prediction process, making them not fully scalable to high-dimensional monotonic features. As the network size increases, the computational cost of the solvers for both methods becomes prohibitively high.

Table 1 presents a comparison of different monotonic neural network methods and their respective capabilities. While some existing methods demonstrate competence in certain aspects, others fall short. While addressing the limitations in learning monotonic neural networks, it is important to prioritize scalability concerning the increasing number of monotonic inputs and network size, alongside achieving good model performance. To address these challenges, this research proposes a new learning approach. Specifically, our study introduces a simple but novel architecture of the partially connected network which incorporates a 'scalable monotonic hidden layer' comprising three units: the exponentiated unit, ReLU unit, and confluence unit. This allows for the repetitive integration of the scalable monotonic hidden layers without other structural constraints. Consequently, our method offers ease of implementation and rapid training through the conventional error-backpropagation

Table 1: Comparison of monotonic neural networks with their capabilities

| Method | end-to-end learning | guaranteeing monotonicity | scalable | trainable using traditional gradient descent |
|---|---|---|---|---|
| Approaches with Regularization | | | | |
| Certified MNN (Liu et al., 2020) | ✗ | ✗ | ✗ | ✓ |
| COMET (Sivaraman et al., 2020) | ✗ | ✓ | ✗ | ✓ |
| Approaches with Customized Architectures | | | | |
| HLL (Yanagisawa et al., 2022) | ✓ | ✓ | ✗ | ✓ |
| LMN (Nolte et al., 2022) | ✓ | ✓ | ✓ | ✗ |
| Constrained MNN (Runje & Shankaranarayana, 2023) | ✓ | ✓ | ✓ | ✗ |
| Ours | ✓ | ✓ | ✓ | ✓ |

algorithm. We accordingly term this method as Scalable Monotonic Neural Networks (SMNN). Furthermore, our method enhances function approximation capabilities. Through numerical experiments, we demonstrate the superiority of our approach when compared to the state-of-the-art methods. Additionally, our method effectively incorporates monotonicity as an inductive bias into the model, leading to improved generalization performance. In matters of scalability, even as monotonic inputs and network size expand, our method demonstrates efficient training with minimally required time increments.

## 2    MONOTONICITY AND PARTIAL MONOTONICITY

This section provides the definition of monotonicity for functions, starting with the definition for a univariate function. Subsequently, we extend the definition to cover monotonicity and partial monotonicity for a multivariate function.

**Monotonicity:**    Let $f(x)$ be a continuous and differentiable univariate function, which maps a normalized input $x \in \mathcal{X} = [0, 1]$ to $\mathcal{Y}$ of $\mathbb{R}$. The function $f(x)$ is said to be monotonically non-decreasing iff $x \leq x' \Rightarrow f(x) \leq f(x')$ holds for any two points $x, x' \in \mathcal{X}$. Similarly, it is monotonically non-increasing iff $x \leq x' \Rightarrow f(x) \geq f(x')$ is satisfied. Alternatively, $f(x)$ is monotonically non-decreasing iff $\forall x, \frac{\partial f(x)}{\partial x} \geq 0$ is met, and it is monotonically non-increasing iff $\forall x, \frac{\partial f(x)}{\partial x} \leq 0$ is satisfied.

**Monotonicity and partial monotonicity:**    Let $f(\mathbf{x})$ be a continuous and differentiable multivariate function, which maps a normalized input $\mathbf{x} \in \mathcal{X} = [0, 1]^d$ to $\mathcal{Y}$ of $\mathbb{R}$. Suppose that the input vector $\mathbf{x}$ is partitioned into $\mathbf{x} = (\mathbf{x}_m, \mathbf{x}_{\neg m}) \in \mathbb{R}^m \times \mathbb{R}^{d-m}$, where $0 < m < d$. The function $f(\mathbf{x})$ is said to be partially monotonically non-decreasing on $\mathbf{x}_m$, which $i$th element is $x_i$, iff $\forall i, \ x_i \leq x_i', \ \forall j \neq i, \ x_j = x_j' \Rightarrow f(\mathbf{x}) \leq f(\mathbf{x}')$ holds for any two points $\mathbf{x}, \mathbf{x}' \in \mathcal{X}$. Similarly, it is monotonically non-increasing iff $\forall i, \ x_i \leq x_i', \ \forall j \neq i, \ x_j = x_j' \Rightarrow f(\mathbf{x}) \geq f(\mathbf{x}')$ is satisfied. Alternatively, $f(\mathbf{x})$ is partially monotonically non-decreasing for $\mathbf{x}$ iff $\forall x_i \in \mathbf{x}_m, \frac{\partial f(\mathbf{x})}{\partial x_i} \geq 0$ is met, and it is partially monotonically non-increasing iff $\forall x_i \in \mathbf{x}_m, \frac{\partial f(\mathbf{x})}{\partial x_i} \leq 0$ is satisfied.

As observed, $\mathbf{x}_m$ represents monotonic features while $\mathbf{x}_{\neg m}$ signifies non-monotonic features. Simultaneously, $m$ denotes the number of monotonic features. In the multivariate case, when $m = d$, partial monotonicity can be treated as monotonicity. Within the neural network training context, the function $f$ can be interpreted as a neural network.

## 3    SCALABLE MONOTONIC NEURAL NETWORK

In this section, we introduce our proposed SMNN method. Our method incorporates three key elements: the utilization of exponentiated weights to ensure monotonicity, the adoption of ReLU-$n$ activation for universal function approximation, and a partially connected network structure that preserves monotonicity for monotonic inputs while capturing interactions among all input features. Because of these characteristics, our method achieves scalability by eliminating the need for any pre- or post-processing steps. Instead, it simply employs conventional error-backpropagation to train a neural network efficiently.

**Exponentiated weights:** Exponentiated formulations have been widely used in various machine learning studies for different purposes. For example, Neural Additive Models (NAM) (Agarwal et al., 2021) introduced the *Exu-centered layer* to increase the flexibility of univariate sub-networks. The use of exponentiated weights in this layer allows even small changes in the input to result in significant changes in the output within a specified range. In the FLOW-based generation model REAL-NVP (Dinh et al., 2016), exponentiated formulations were employed to ensure that the determinant of the Jacobian matrix of the affine coupling layer remains positive. In line with these previous works, but with a different purpose, we propose a hidden node that utilizes exponentiated weights for monotonic input features, as defined by

$$h_{exp}(\mathbf{x}_m) = \sigma(\exp(\mathbf{w})^\top \mathbf{x}_m + b) \tag{1}$$

where $\sigma$ is an activation function and $b$ is a bias. It becomes evident that when the activation function is monotonically non-decreasing, the hidden node preserves its monotonic behavior. Therefore, we exploit the ability of exponentiated weights to preserve the monotonicity of the hidden node, making them a good choice for learning monotonic neural networks. While the concept is similar to an early study (Zhang & Zhang, 1999), our research is distinguished by the use of a different activation function and a specially designed network structure. These will be described in detail below.

**Activation functions:** In modern deep learning, ReLU is commonly used as an activation function due to its advantages such as alleviating gradient vanishing (Nair & Hinton, 2010; He et al., 2016). Another benefit of this activation function is that it allows neural network training to be formulated as an integer programming problem (Liu et al., 2020) or other optimization problems (Sivaraman et al., 2020), simplifying the training process. However, it is known that a ReLU network with all weights constrained to be positive is either convex or concave, depending on the direction of ReLU (Liu et al., 2020). This means that the neural network lacks the universal approximation property. To overcome this limitation, we employ the ReLU-$n$ activation function (Liew et al., 2016), which is given by

$$\text{ReLU-}n(x) = \begin{cases} n & \text{if } x > n, \ x \to n^+, \\ x & \text{if } 0 < x < n, \ x \to n^-, \ x \to 0^+, \\ 0 & \text{if } x < 0, \ x \to 0^-. \end{cases} \tag{2}$$

A previous research has demonstrated that neural networks with positive-constrained weights and a threshold activation function are universal approximators of partially monotonic functions with at least a depth of 4 hidden layers (Mikulincer & Reichman, 2022). The nature of the ReLU-$n$ function in equation 2 is a piecewise-linear function with a threshold at 0 and $n$. Unlike ReLU, this function exhibits both convexity at zero and concavity at $n$. Importantly, it is a monotonically non-decreasing function, thus ensuring the preservation of monotonicity of the hidden node in equation 1. Alternative activation functions offering the similar benefit, including both convexity and concavity, can be substituted for the ReLU-$n$ function (See our ablation study in Appendix C.1).

**Network structure:** Prior to introducing the proposed network structure, we define three types of hidden nodes. The exponentiated hidden node, denoted in equation 3, is designed for monotonic features ($\mathbf{x}_m$) based on equation 1 and equation 2. For non-monotonic features ($\mathbf{x}_{\neg m}$), the well-known ReLU hidden node, which is given in equation 4, is employed. Lastly, the confluence hidden node is defined in equation 5. Each hidden node forms an individual hidden unit comprising multiple nodes, resulting in three types of hidden units, namely the *exponentiated unit*, ReLU *unit*, and *confluence unit*, as also defined in equation 3, equation 4, and equation 5 respectively. To maintain a consistent activation function throughout the network, it is acceptable to use the ReLU-$n$ function for the ReLU unit (See Appendix C.1). It is worth noting that the hidden nodes in equations 3, 4, and 5 represent the nodes in the first hidden layer, which receive the inputs ($\mathbf{x}_m$, $\mathbf{x}_{\neg m}$). As we progress to the next hidden layer, they receive the outputs from the preceding hidden nodes.

$$\text{Exponentiated unit} := \{h_{exp}\}, \text{ where } h_{exp}(\mathbf{x}_m) = \text{ReLU-}n(\exp(\mathbf{w})^\top \mathbf{x}_m + b) \tag{3}$$

$$\text{ReLU unit} := \{h_{relu}\}, \text{ where } h_{relu}(\mathbf{x}_{\neg m}) = \text{ReLU}(\mathbf{w}^\top \mathbf{x}_{\neg m} + b) \tag{4}$$

$$\text{Confluence unit} := \{h_{conf}\}, \text{ where } h_{conf}(\mathbf{x}_{\neg m}) = \text{ReLU-}n(\mathbf{w}^\top \mathbf{x}_{\neg m} + b) \tag{5}$$

Based on these units, we present our proposed SMNN structure, where each hidden layer comprises the three units. We prepared Fig. 1 illustrating a network with three hidden layers as an example to

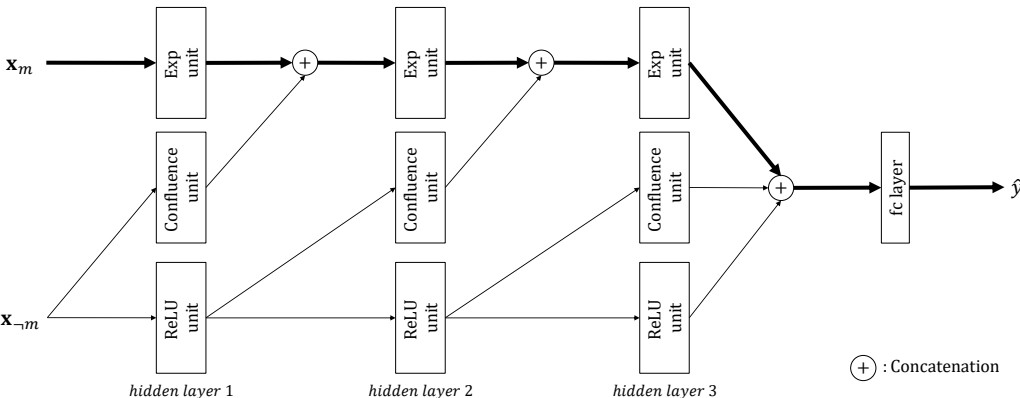

Figure 1: **SMNN Structure.** The features are divided into two groups based on whether each variable requires monotonicity ($\mathbf{x}_m$) or not ($\mathbf{x}_{\neg m}$). The monotonic features are directed exclusively through the *exponentiated unit* and *fully connected layer* to ensure monotonicity, as indicated by the bold line.

help readers better understand. Let

$$l_i = (\{h_{exp}^{(i)}\}, \{h_{conf}^{(i)}\}, \{h_{relu}^{(i)}\}) \tag{6}$$

be the $i$-th hidden layer, which we named as scalable monotonic hidden layer. The units in this layer serve specific roles: the exponentiated unit captures interactions among monotonic features ($\mathbf{x}_m$) while preserving monotonicity, the ReLU unit captures interactions among non-monotonic features ($\mathbf{x}_{\neg m}$), and the confluence unit combines the outputs of the exponentiated and ReLU units to capture interactions between both types of features (See Appendix C.3 for the justification of the confluence unit.). The purpose of using ReLU-$n$ activation functions in the confluence unit nodes is to align the output magnitudes of these nodes with the output magnitudes of the exponentiated unit nodes in the preceding layer. When $i = 1$,

$$\{h_{exp}^{(1)}\} = \{h_{exp}(\mathbf{x}_m)\}, \ \ \{h_{conf}^{(1)}\} = \{h_{conf}(\mathbf{x}_{\neg m})\}, \ \ \{h_{relu}^{(1)}\} = \{h_{relu}(\mathbf{x}_{\neg m})\}. \tag{7}$$

When $i \geq 2$,

$$\begin{aligned}
\{h_{exp}^{(i)}\} &= \{h_{exp}([\{h_{exp}^{(i-1)}\}, \{h_{conf}^{(i-1)}\}])\}, \\
\{h_{conf}^{(i)}\} &= \{h_{conf}(\{h_{relu}^{(i-1)}\})\}, \\
\{h_{relu}^{(i)}\} &= \{h_{relu}(\{h_{relu}^{(i-1)}\})\},
\end{aligned} \tag{8}$$

where $[\cdot, \cdot]$ implies the concatenation of two vectors. Let $\mathbf{h} = l_n$, which is the output vector at the $n$-th hidden layer, and $f(\mathbf{x})$ be the output node in a fully connected output layer. Then,

$$f(\mathbf{x}) = \exp(\mathbf{w}_{n+1})^\top \mathbf{h} + b, \tag{9}$$

where $\mathbf{w}_{n+1}$ is the vector of weights between the $n$-th hidden layer and the output layer. The output node also employs exponentiated weights to ensure monotonicity, without an activation function. It is important to note that while we present the proposed SMNN for a single output node, this expression can be easily extended to accommodate multiple output nodes. Notably, the monotonic features follow a distinct path through the exponentiated unit and the fully connected layer, depicted by the bold line in Fig. 1, ensuring the monotonicity of the proposed SMNN.

**Monotonicity of SMNN:**   We now prove the monotonicity of SMNN by following the monotonicity definition outlined in Section 2. This involves taking the partial derivative of SMNN with respect to a monotonic feature and showing that the derivative is positive. It is worth noting that SMNN is inherently a monotonically non-decreasing function, and for a monotonically non-increasing feature, we can simply multiply the feature by $-1$ and incorporate it into SMNN to maintain its property of monotonic non-increase.

**Theorem 1.** Let $f(\mathbf{x})$ be the proposed SMNN with $n$ scalable monotonic hidden layers and $\mathbf{x}_m = (x_1, \ldots, x_p)$ be a vector of $p$ monotonic input features. For $x_i, i \in \{1, \ldots, p\}$,

$$\frac{\partial f(\mathbf{x})}{\partial x_i} \geq 0.$$

*Proof.* As $x_i$ follows a distinct path through the exponentiated unit and the fully connected layer and we aim to find the partial derivative of $f(\mathbf{x})$ with respect to $x_i$, without loss of generality, we consider a continuous function $g(x)$ given by

$$g(x) = \exp(w_{n+1})t_n + b_{n+1},$$

where

$$t_i = \text{ReLU-}n(\exp(w_i)t_{i-1} + b_i), \ i = 2, \ldots, n,$$

and

$$t_1 = \text{ReLU-}n(\exp(w_1)x + b_1).$$

By showing $\frac{dg(x)}{dx} \geq 0$, we can show $\frac{\partial f(\mathbf{x})}{\partial x_i} \geq 0$. It is quite straightforward to illustrate $\frac{dg(x)}{dx} \geq 0$. We first describe the derivative of ReLU-$n$(s) function as follows:

$$R' = \text{ReLU-}n'(s) \to \begin{cases} 0 & \text{if } s > n, \ s \to n^+, \\ 1 & \text{if } 0 < s < n, \ s \to n^-, \ s \to 0^+, \\ 0 & \text{if } s < 0, \ s \to 0^-. \end{cases}$$

Then,

$$\frac{dt_1}{dx} = R' \cdot \exp(w_1) \geq 0.$$

Likewise,

$$\frac{dt_2}{dx} = R' \cdot \exp(w_2) \cdot \frac{dt_1}{dx} = R' \cdot \exp(w_2) \cdot R' \cdot \exp(w_1) \geq 0,$$

and

$$\frac{dt_n}{dx} = R' \cdot \exp(w_n) \cdot \frac{dt_{n-1}}{dx} = R' \cdot \exp(w_n) \cdot R' \cdot \exp(w_{n-1}) \cdots R' \cdot \exp(w_1) \geq 0.$$

We finally show that

$$\frac{dg(x)}{dx} = \exp(w_{n+1}) \cdot \frac{dt_n}{dx} \geq 0.$$

$\square$

## 4 Numerical Experiments

### 4.1 Experiments on synthetic datasets

We first assessed the effectiveness of our proposed method by conducting experiments on synthetic datasets. In addition to verifying the monotonicity aspect, we thoroughly examined the scalability and generalization performance of our method.

**Scalability test:** In this experiment, we used two synthetic datasets to evaluate the scalability with respect to the network size and the number of monotonic features. For the first aspect of the scalability, we used a two-dimensional example obtained from (Liu et al., 2020). The function is defined in equation 10, where monotonicity is imposed on $y$ only between $x$ and $y$. We created a dataset consisting of 1000 instances with $x$ and $y$ values generated from a uniform distribution within the range of 0 and 1. The dataset was divided into a 70% training set and a 30% test set.

$$f(x, y) = a \sin(\frac{25x}{\pi}) + b (x - 0.5)^3 + c \exp(y) + y^2, \tag{10}$$

$$a, b, c \in \{0.3, 0.6, 1.0\}, \ x, y \in [0, 1]$$

Fig. 2(a) demonstrates that our proposed method (with two scalable monotonic hidden layers) accurately fit the original function, ensuring monotonic increase of the function with respect to $y$.

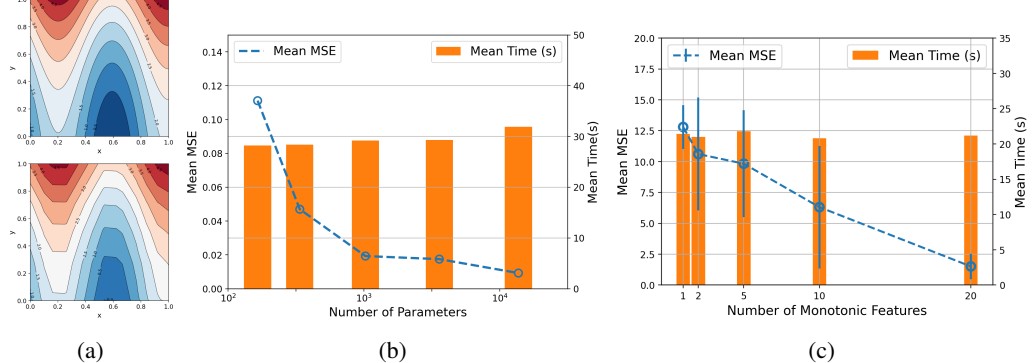

Figure 2: (a) **Upper:** Contour plot of original function with $a, b, c = 1$ and $x, y \in [0, 1]$. **Lower:** Contour plot of the fitted model by our method (MSE=0.0007). (b) Scalability test with respect to increasing network size: We tested the proposed method for all combinations of $a, b, c$. Blue line represents the average MSE from all 27 runs. Orange bar indicates the average training time required for all 27 runs. (c) Scalability test with respect to increasing number of monotonic features: Blue line represents the average MSE from five repetitions; Orange bar indicates the average time required for five repetitions.

Furthermore, Fig. 2(b) illustrates that as the number of parameters increases, the average mean-squared error (MSE) of the model decreases while the computation time remains nearly constant. This indicates the high scalability and extensibility of our proposed method to accommodate larger networks.

In addition, to investigate the scalability with respect to the number of monotonic features, we designed a 40-dimensional dataset, excluding the output feature, comprising 20 uniform distribution-based dummy inputs and 20 monotonic inputs determined by equation 11.

$$f(\mathbf{x}) = \sum_{i=1}^{5} \alpha_i x_i + \sum_{i=6}^{10} \alpha_i \exp x_i + \sum_{i=11}^{15} \alpha_i (x_i + \sin x_i) + \sum_{i=16}^{20} \alpha_i (x_i + \cos x_i) + 20\epsilon, \quad (11)$$

$$x_i \in [0, 1] \ (i = 1, \dots, 20), \ \epsilon \sim N(0, 1)$$

$$\{\alpha_i\} = \{5, 4, 3, 6, 8, 5, 4, 3, 2, 1, 5, 4, 3, 2, 1, 5, 4, 3, 2, 1\} (i = 1, \dots, 20)$$

We generated 1000 instances, allocating 700 instances for training and 300 for the test set. The study involved systematically increasing $m$ from 1 to 20. It is worth noting that a monotonic input either belonged to $\mathbf{x}_m$ or $\mathbf{x}_{\neg m}$. Fig. 2(c) illustrates that as the number of monotonic inputs increased from 1 to 20, SMNN exhibited consistent training times (at around 20 seconds), while decreasing its MSE performance. See Appendix C.4 for the case of 200 monotonic features.

**Generalization performance test:** We conducted a series of experimental evaluations to assess the generalization performance of the proposed SMNN method with guaranteed monotonicity, using the well-known Friedman function [1] (Friedman, 1991). To introduce variability into the original function, Gaussian noise $\epsilon \sim N(0, 1)$ was incorporated, and the scaling factor $\lambda$ applied to the noise was gradually increased. The Friedman model, defined in equation 12, imposes monotonicity on $x_4$ and $x_5$.

$$f(\mathbf{x}) = 10 \sin(\pi x_1 x_2) + 20(x_3 - 0.5)^2 + 10 x_4 + 5 x_5 + \lambda \epsilon, \quad (12)$$

$$x_i \in [0, 1] \ (i = 1, ..., 5), \ \epsilon \sim N(0, 1)$$

In our analysis, we compared MSE across three different networks: the proposed SMNN with monotonicity enforced on $x_4$ and $x_5$, a structurally identical network with SMNN but using only ReLU activation functions (without exponentiated weights and ReLU-$n$), and a typical Multi-Layer Perceptron (MLP) with matching layer depth and node count. To ensure robust evaluation, we employed a five-fold cross-validation strategy, using 80% of the data for training and 20% for testing.

[1] https://www.sfu.ca/~ssurjano/fried.html

For each folding, we conducted five independent runs, which resulted in a total of 25 experimental trials.

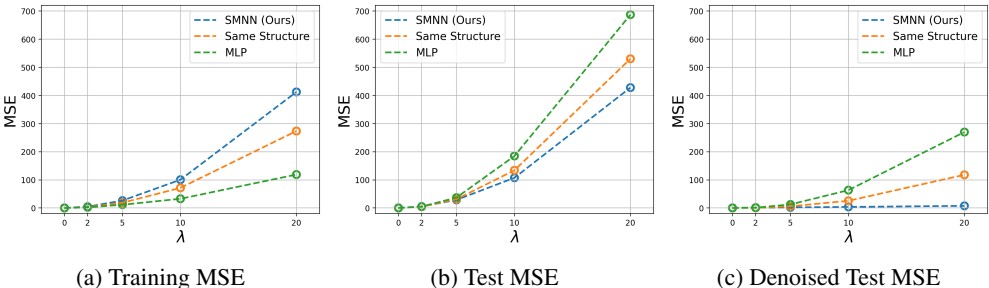

(a) Training MSE          (b) Test MSE          (c) Denoised Test MSE

Figure 3: Results from the Friedman function experiments

The experimental results are presented in Fig. 3 (For more detailed results, please refer to Appendix B). The training MSEs are shown in Fig. 3(a), where SMNN exhibited the highest error, while MLP achieved the lowest. This suggests that the imposition of monotonicity constraints significantly affected the learning dynamics of SMNN. Conversely, in terms of the testing MSE shown in Fig. 3(b), MLP had the highest error, followed by the structurally identical network, while SMNN yielded the lowest MSE. For MLP, the testing MSE was nearly seven times greater than the training MSE when $\lambda$ reached 20, indicating overfitting to the noise. In contrast, the testing MSE of SMNN remained consistent with the training MSE across different $\lambda$ values, indicating robustness to noise. Notably, Fig. 3(c) presents an interesting observation where the noise of the test dataset was eliminated. In this scenario, the MSE of SMNN becomed significantly smaller than that of the other networks. These results suggest that incorporating monotonicity as an appropriate inductive bias enhanced the generalization performance of the network, as demonstrated in our experimental findings. Therefore, we conclude from this experiment that imposing monotonicity could improve generalization performance.

## 4.2 Experiments on real-world datasets

Having established the validity of our method in the previous subsection4.1, we proceed to compare it with the state-of-the-art methods discussed earlier. We conducted both regression and classification tasks using several publicly accessible datasets that were used in previous studies. The Auto-MPG[2] and Blog Feedback (Spiliopoulou et al., 2014) datasets were used for regression tasks, while the COMPAS (Angwin et al., 2016), Heart Disease[3], and Loan Defaulter[4] datasets were employed for classification. Similar to the synthetic data experiments, we applied a 5-fold cross-validation strategy, performing 25 runs for each dataset. Appendix A provides details regarding the data description and training configurations. The experimental results are presented in Tables 2 and 3 respectively. The reported metrics include model complexity (number of parameters) if available, and prediction performances such as MSE and RMSE for regression and accuracy for classification, with mean and standard deviation obtained from repetitions. The benchmarking methods included in the tables for comparison are Isotonic in (Kalai & Sastry, 2009), XGBoost in (Chen et al., 2015), Crystal in (Milani Fard et al., 2016), DLN in (You et al., 2017), Min-Max Net in (Daniels & Velikova, 2010), Non-Neg-DNN in (Liu et al., 2020), Certified MNN in (Liu et al., 2020), COMET in (Sivaraman et al., 2020), LMN in (Nolte et al., 2022), and Constrained MNN in (Runje & Shankaranarayana, 2023).

Tables 2 and 3 present comprehensive performance evaluations of the proposed methodology with the comparative techniques. Table 2 shows the experimental results and model complexities with the dataset featured in the previous studies (Liu et al., 2020; Nolte et al., 2022; Runje & Shankaranarayana, 2023). Our proposed approach demonstrated superior performance on both the COMPAS and Blog Feedback datasets in comparison to existing methodologies. In the case of the Loan Defaulter dataset, while our method did not achieve the top position, it showed comparable performance to recently

---

[2] `https://archive.ics.uci.edu/ml/datasets/auto+mpg`
[3] `https://archive.ics.uci.edu/ml/datasets/Heart+Disease`
[4] `https://www.kaggle.com/wendykan/lenging-club-loan-data`

Table 2: Comparison of our method with other methods described in (Liu et al., 2020; Nolte et al., 2022; Runje & Shankaranarayana, 2023) (" † " indicates the statistical tie with the best.)

| Method | COMPAS | | Blog Feedback | | Loan Defaulter | |
|---|---|---|---|---|---|---|
| | Parameters | Test Acc ↑ | Parameters | RMSE ↓ | Parameters | Test Acc ↑ |
| Isotonic | N.A. | 67.6% | N.A. | 0.203 | N.A. | 62.1% |
| XGBoost | N.A. | $(68.5 \pm 0.1)\%$ | N.A. | $0.176 \pm 0.005$ | N.A. | $(63.7 \pm 0.1)\%$ |
| Crystal | 25840 | $(66.3 \pm 0.1)\%$ | 15840 | $0.164 \pm 0.002$ | 16940 | $(65.0 \pm 0.1)\%$ |
| DLN | 31403 | $(67.9 \pm 0.3)\%$ | 27903 | $0.161 \pm 0.001$ | 29949 | $(65.1 \pm 0.2)\%$ |
| Min-Max Net | 42000 | $(67.8 \pm 0.1)\%$ | 27700 | $0.163 \pm 0.001$ | 29000 | $(64.9 \pm 0.1)\%$ |
| Non-Neg-DNN | 23112 | $(67.3 \pm 0.9)\%$ | 8492 | $0.168 \pm 0.001$ | 8502 | $(65.1 \pm 0.1)\%$ |
| Certified MNN | 23112 | $(68.8 \pm 0.9)\%†$ | 8492 | $0.158 \pm 0.001$ | 8502 | $(65.2 \pm 0.1)\%$ |
| LMN | 37 | $(\mathbf{69.3 \pm 0.1})\%$ | 2225 | $0.160 \pm 0.001$ | 753 | $(\mathbf{65.4 \pm 0.0})\%$ |
| Constrained MNN | 2317 | $(69.2 \pm 0.2)\%†$ | 1101 | $0.156 \pm 0.001$ | 177 | $(65.3 \pm 0.1)\%†$ |
| SMNN (Ours) | 2657 | $(\mathbf{69.3 \pm 0.9})\%$ | 1421 | $\mathbf{0.150 \pm 0.001}$ | 501 | $(65.0 \pm 0.1)\%$ |

Table 3: Comparison of our method with other methods described in (Sivaraman et al., 2020; Nolte et al., 2022; Runje & Shankaranarayana, 2023) (" † " indicates the statistical tie with the best.)

| Method | Auto MPG | Heart Disease |
|---|---|---|
| | MSE ↓ | Test Acc ↑ |
| Min-Max Net | $10.14 \pm 1.54$ | $0.75 \pm 0.04$ |
| DLN | $13.34 \pm 2.42$ | $0.86 \pm 0.02$ |
| COMET | $8.81 \pm 1.81$ | $0.86 \pm 0.03$ |
| LMN | $7.58 \pm 1.20†$ | $\mathbf{0.90 \pm 0.02}$ |
| Constrained MNN | $8.37 \pm 0.08$ | $0.89 \pm 0.00†$ |
| SMNN (Ours) | $\mathbf{7.44 \pm 1.20}$ | $0.88 \pm 0.04†$ |

proposed methods, with a relatively small network size. Table 3 also includes the experimental results obtained from the previous works (Sivaraman et al., 2020; Nolte et al., 2022; Runje & Shankaranarayana, 2023). In the context of the Auto-MPG dataset, our proposed method and LMN outperformed all others in the comparison, while, for the Heart Disease dataset, our approach did not achieve the level of performance attained by LMN, which secured the top position. When it came to the classification tasks, our method achieved the top result in a specific case, while in other cases, it performed comparably to the benchmarks.

## 5 CONCLUSIONS

In this study, we introduced SMNN, a simple but novel learning method designed to ensure the monotonicity of trained models. Our approach combines the exponentiated unit, ReLU unit, and confluence unit in a hidden layer to achieve both monotonicity and effective function approximation. Being an end-to-end learning approach, our method exhibits scalability with respect to both network size and number of monotonic features. Through extensive experiments on synthetic and real-world datasets, we demonstrated the scalability and comparable performance of our method against other benchmarking methods. For future work, we will focus on mathematically proving the benefits of incorporating monotonicity as an inductive bias in neural networks to enhance their generalization performance. Additionally, we plan to apply SMNN to a broader range of real-world applications that require both monotonicity and scalability.

## 6 REPRODUCIBILITY

To facilitate reproducibility, we have provided sufficiently detailed descriptions of the implementation of the proposed method both in the main text and appendices. In particular, in Section 3, we clearly present the complete structure of our proposed method along with its components. Appendix A provides detailed experimental descriptions and training configurations. All implemented code can be found in the supplements, which is also available at `https://github.com/retna319/SMNN`. Experiments were conducted using the provided code.

ACKNOWLEDGMENTS

This work was supported in part by the National Research Foundation of Korea(NRF) funded by the Korean Government (Ministry of Science and ICT (MSIT)) under Grant 2020R1A5A1019 649 and in part by the Institute for Information and Communications Technology Planning & Evaluation (IITP) funded by the Korean Government (MSIT) under Grant 20210002920012002.

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

# A    DETAILED EXPERIMENT DESCRIPTIONS

## A.1    TRAINING CONFIGURATIONS

All experiments were conducted on a system equipped with an Intel(R) Core(TM) i7-8700 CPU @ 3.20GHz processor and 32.0GB of DDR3 RAM, running the Windows 10 operating system. The experiments were implemented in Python, utilizing the PyTorch library[5] (version 1.10.2). Stochastic optimization for training networks was performed using the Adam optimizer. The best hyperparameters were determined through grid search, considering different batch sizes (128, 256, 512), learning rates (0.05, 0.005, 0.002, 0.0005). The neural network architecture consisted of two scalable monotonic hidden layers. We consistently set the hyperparameter $n$ in the ReLU-$n$ activation function to 1 throughout all our experiments, because the primary purpose with this activation function was to utilize the convex and concave properties, making detailed tuning of this value unnecessary. For regression tasks, the mean squared error loss was employed, while for classification tasks, the cross-entropy loss was used. The number of epochs was set to either 1000 or 500 considering datasets.

## A.2    DESCRIPTIONS FOR REAL-WORLD DATASETS

Table 4 provides the detailed descriptions for the real-world datasets used in our experiments. It includes the tasks, number of instances, number of features, number of monotonic features, and names of the monotonic features. Monotonically non-increasing features are denoted in bold, while monotonically non-decreasing features are non-bold. For monotonically non-increasing features, we multiplied the features by $-1$ to ensure monotonic non-decrease. Additionally, we normalized all inputs to a 0-1 scale. The best hyperparameter configurations for each dataset are presented in Table 5. This table shows the network structure with two scalable monotonic layers, as previously mentioned. The numbers within parentheses represent the node counts for each exponentiated unit, confluence unit, and ReLU unit.

Table 4: Descriptions for real-world datasets

| Dataset | Task | # Instances | # Features | # Monotonic features | Monotonic features |
|---|---|---|---|---|---|
| COMPAS | Classification | 6172 | 13 | 4 | number of prior adult convictions, number of juvenile felony, number of juvenile misdemeanor, number of other convictions |
| Heart Disease | Classification | 303 | 13 | 2 | trestbps, chol |
| Loan defaulter | Classification | 488909 | 28 | 5 | number of public record bankruptcies, dept-to-income ratio, **credit score**, **length of employment**, **annual income** |
| Blog feedback | Regression | 54270 | 276 | 8 | A51,A52,A53,A54,A56,A57,A58,A59 |
| Auto-MPG | Regression | 398 | 7 | 3 | **weights, displacement, horse power** |

Table 5: Hyperparameters of SMNN for real-world datasets

| Dataset | # Parameters | Network structure | Learning rate | Batch size |
|---|---|---|---|---|
| COMPAS | 2657 | $(32, 16, 16), (32, 16, 16)$ | 0.005 | 128 |
| Heart Disease | 7905 | $(64, 16, 32), (64, 16, 32)$ | 0.002 | 128 |
| Loan Defaulter | 501 | $(16, 4, 4), (8, 2, 4)$ | 0.0005 | 512 |
| Blog Feedback | 1421 | $(16, 2, 2), (8, 2, 4)$ | 0.0005 | 256 |
| Auto-MPG | 14193 | $(128, 16, 64), (64, 16, 32)$ | 0.005 | 128 |

**COMPAS:**    The COMPAS dataset is a classification dataset containing the criminal records in Florida. The task is to classify whether an arrested individual will reoffend or not within two years.

---

[5]https://pytorch.org/

This dataset consists of 13 features, with four of them considered as monotonic features. Specifically, the following features exhibit monotonic non-decreasing behavior: *number of prior adult convictions*, *number of juvenile felony*, *number of juvenile misdemeanor* and *number of other convictions*. It is worth mentioning that this dataset has been used in several published papers (Angwin et al., 2016; Rudin et al., 2020), although with a minor ethical concern. While a few individuals perceive it as possibly problematic, the majority do not share the same concern.

**Heart Disease:**  The Heart Disease dataset is used to classify whether a person has heart disease or not. The presence of heart disease monotonically increases with respect to *trestbps* (resting blood pressure) and *chol* (cholesterol level).

**Loan Defaulter:**  The Loan Defaulter dataset includes complete loan data issued between 2007 and 2015 and addresses a classification task aimed at predicting loan defaulters using 28 features. Among these 28 features, five exhibit monotonic behavior. Specifically, the *number of public record bankruptcies* and *dept-to-income ratio* are monotonically non-decreasing, while the **credit score**, **length of employment**, and **annual income** are monotonically non-increasing.

**Blog Feedback:**  The Blog Feedback dataset is a regression dataset used to predict the number of comments within 24 hours. It comprises 54,270 blog post records and includes 276 features. Eight of these features (A51, A52, A53, A54, A56, A57, A58, A59) exhibit monotonically non-decreasing behavior.

**Auto-MPG:**  The Auto-MPG dataset is a regression dataset used to predict miles per gallon (*mpg*). It consists of 7 features, and among them, **weights**, **displacement**, and **horse power** need to be enforced as monotonically non-increasing.

## A.3 WEIGHT INITIALIZATION AND UPDATE STRATEGY FOR SMNN

In the proposed SMNN method, the exponentiated unit incorporates the exponential value of weights and utilizes the ReLU-$n$ activation function, which has a threshold at 0 and $n$. This design can lead to node inactivation when the activation function's gradient becomes zero at high input values due to the influence of the exponentiated weights. To mitigate this potential risk, we have implemented specific weight initialization and weight update strategies. Regarding the initialization strategy, instead of generating initial weights from the typical Gaussian distribution, we opted for a uniform distribution spanning from a negative lower limit to a small positive upper limit, such as $uniform(-20, 2)$. Consequently, the exponentiated values of these weights are distributed in proximity to zero, creating a skewed long-tail distribution. This distribution shape effectively prevents the occurrence of zero gradients in the activation function. Concerning the update strategy, we intentionally set the gradient of inactivated nodes to a very small non-zero value, such as $0.01$. By ensuring a non-zero gradient, we enable the exploration of alternative optimal weight configurations and minimize the risk of getting trapped in local optima. These strategies are designed to enhance the search for better optimal weights by mitigating the possibility of node inactivation caused by large input values and zero gradients.

# B SUPPLEMENTAL REPORT OF EXPERIMENTAL RESULTS

## B.1 TABULATED NUMERICAL RESULTS FOR FIGURE 2(B)

Figure 2(b) was created using the data in the following table.

Table 6: Mean MSE at varying number of parameters

| Network | # Parameters | Test MSE | Time (s) |
|---------|-------------|----------|----------|
| SMNN (*size* 1) | 165 | $0.111 \pm 0.147$ | $28.220 \pm 1.219$ |
| SMNN (*size* 2) | 337 | $0.047 \pm 0.112$ | $28.370 \pm 1.114$ |
| SMNN (*size* 3) | 1025 | $0.019 \pm 0.038$ | $29.180 \pm 2.019$ |
| SMNN (*size* 4) | 3585 | $0.018 \pm 0.050$ | $29.296 \pm 1.170$ |
| SMNN (*size* 5) | 13697 | $0.009 \pm 0.016$ | $31.889 \pm 1.948$ |

## B.2 TABULATED NUMERICAL RESULTS FOR FIGURE 2(C)

Figure 2(c) was created using the data in the following table.

Table 7: Mean MSE at varying number of monotonic features

| Network | # Monotonic Features | Test MSE | Time (s) |
|---------|---------------------|----------|----------|
| SMNN ($m = 1$) | $\{x_{13}\}$ | $12.80 \pm 1.78$ | $21.4 \pm 1.14$ |
| SMNN ($m = 2$) | $\{x_5, x_{13}\}$ | $10.61 \pm 4.57$ | $21.0 \pm 0.00$ |
| SMNN ($m = 5$) | $\{x_1, x_5, x_9, x_{13}, x_{17}\}$ | $9.83 \pm 4.32$ | $21.8 \pm 1.09$ |
| SMNN ($m = 10$) | $\{x_1, x_3, x_5, x_7, x_9, x_{11}, x_{13}, x_{15}, x_{17}, x_{19}\}$ | $6.31 \pm 4.96$ | $20.8 \pm 0.44$ |
| SMNN ($m = 20$) | $\{x_1, ..., x_{20}\}$ | $1.51 \pm 1.03$ | $21.2 \pm 1.64$ |

## B.3 TABULATED NUMERICAL RESULTS FOR FIGURE 3

Figure 3 was created using the data in the following table.

Table 8: Results from the Friedman function experiments

| Network | Training MSE | Test MSE | Denoised Test MSE |
|---------|-------------|----------|-------------------|
| SMNN ($\lambda = 0$) | $0.68 \pm 0.42$ | $0.78 \pm 0.50$ | $0.78 \pm 0.50$ |
| SMNN ($\lambda = 2$) | $4.85 \pm 0.23$ | $5.42 \pm 0.40$ | $1.25 \pm 0.28$ |
| SMNN ($\lambda = 5$) | $26.53 \pm 0.65$ | $28.89 \pm 3.05$ | $2.22 \pm 0.47$ |
| SMNN ($\lambda = 10$) | $100.64 \pm 1.73$ | $107.46 \pm 6.45$ | $4.02 \pm 0.79$ |
| SMNN ($\lambda = 20$) | $412.44 \pm 12.98$ | $427.92 \pm 41.84$ | $7.73 \pm 1.55$ |
| Same Structure ($\lambda = 0$) | $0.02 \pm 0.01$ | $0.03 \pm 0.02$ | $0.03 \pm 0.02$ |
| Same Structure ($\lambda = 2$) | $3.27 \pm 0.17$ | $4.63 \pm 0.41$ | $0.78 \pm 0.24$ |
| Same Structure ($\lambda = 5$) | $18.81 \pm 1.15$ | $31.46 \pm 2.04$ | $6.03 \pm 1.16$ |
| Same Structure ($\lambda = 10$) | $71.69 \pm 6.68$ | $134.16 \pm 10.80$ | $25.69 \pm 5.83$ |
| Same Structure ($\lambda = 20$) | $273.58 \pm 27.75$ | $530.27 \pm 60.01$ | $118.02 \pm 28.31$ |
| MLP ($\lambda = 0$) | $0.02 \pm 0.01$ | $0.04 \pm 0.02$ | $0.04 \pm 0.02$ |
| MLP ($\lambda = 2$) | $2.53 \pm 0.29$ | $5.18 \pm 0.46$ | $1.28 \pm 0.27$ |
| MLP ($\lambda = 5$) | $11.77 \pm 2.13$ | $37.01 \pm 4.10$ | $12.73 \pm 2.45$ |
| MLP ($\lambda = 10$) | $32.92 \pm 8.60$ | $184.95 \pm 21.64$ | $63.61 \pm 9.63$ |
| MLP ($\lambda = 20$) | $118.95 \pm 30.99$ | $687.04 \pm 74.25$ | $270.01 \pm 42.59$ |

## C ABLATION STUDIES

### C.1 DIFFERENT ACTIVATION SETTINGS

In our experiments, we applied the ReLU-1 function to both the exponentiated and confluence units, and the standard ReLU function to the ReLU unit (denoted by 'Proposed (Ours)' in Table 9). We believe that it is acceptable to use the ReLU-1 function for the ReLU unit to maintain a consistent activation function throughout the network (denoted by 'ReLU-1' in Table 9). Since the ReLU-$n$ function offers convexity and concavity, other activation functions providing the similar benefit can replace ReLU-$n$. In this ablation study, we explored the use of the sigmoid and hyperbolic tangent functions for all three units (identified as 'Sigmoid' and 'Tanh', respectively, in Table 9). Based on the results, we cannot assert performance differences among the activation functions.

Table 9: Performance comparison among different activation settings ("†" indicates the statistical tie with the best.)

| Method | Auto MPG | Heart Disease |
| --- | --- | --- |
| | MSE $\downarrow$ | Test Acc $\uparrow$ |
| Proposed (Ours) | $7.44 \pm 1.20$† | $\mathbf{0.88 \pm 0.04}$ |
| ReLU-1 | $\mathbf{7.36 \pm 1.67}$ | $0.85 \pm 0.04$† |
| Sigmoid | $13.96 \pm 3.18$ | $0.85 \pm 0.07$† |
| Tanh | $7.83 \pm 1.45$† | $0.87 \pm 0.05$† |

### C.2 DIFFERENT $n$ VALUES IN RELU-$n$

While an optimal value for the hyperparameter $n$ in ReLU-$n$ may exist for a specific training dataset, to demonstrate its insensitivity, we conducted experiments with varying $n$ based on the Auto MPG and Heart Disease datasets. The results are presented in Table 10, showing no performance differences across different $n$ values.

Table 10: Performance comparison among different $n$ values in ReLU-$n$ ("†" indicates the statistical tie with the best.)

| Method | Auto MPG | Heart Disease |
| --- | --- | --- |
| | MSE $\downarrow$ | Test Acc $\uparrow$ |
| ReLU-1 | $7.44 \pm 1.20$† | $\mathbf{0.88 \pm 0.04}$ |
| ReLU-2 | $7.40 \pm 1.31$† | $0.87 \pm 0.04$† |
| ReLU-6 | $7.38 \pm 1.16$† | $0.87 \pm 0.04$† |
| ReLU-10 | $\mathbf{7.36 \pm 1.39}$ | $0.87 \pm 0.04$† |

### C.3 EFFECT OF CONFLUENCE UNITS

We conducted an ablation study to validate the impact of the confluence units in learning interactions. The function designed for the experiment is defined in equation 13. The output is determined by $x_1$ and $x_2$. To control the strength of the interaction effect, we varied the coefficient $\alpha$ of the interaction term from 0 to 20. We imposed monotonicity on $x_1$ only, even though $x_2$ is monotonic at $[0, 2]$. We generated 1000 instances, allocating 700 instances for training and 300 for the test set. The number of hidden layers for both networks, with and without confluence units, was set to two. However, the number of hidden nodes was assigned differently: $\{(64, 64, 64), (32, 32, 32)\}$ for SMNN with confluence units, $\{(128, 0, 64), (64, 0, 32)\}$ for SMNN without confluence units. The MSE comparison in Table 11 illustrates that with increased interaction effects, more prominent performance differences emerged, consistently favoring SMNN with confluence units. This validates the effectiveness of employing the confluence unit to capture interactions between monotonic and non-monotonic features.

$$f(x_1, x_2) = x_1 + \alpha \cdot x_1^3 \cdot x_2^2 + x_2, \tag{13}$$
$$x_1, x_2 \in [0, 2], \ \alpha \in [0, 1, 2, 5, 10, 20]$$

Table 11: Performance (MSE) comparison between with and without confluence units

| $\alpha$ | with confluence units | without confluence units |
|---|---|---|
| $\alpha = 0$ | $0.46 \pm 0.05$ | $0.96 \pm 0.10$ |
| $\alpha = 1$ | $0.58 \pm 0.05$ | $8.35 \pm 0.20$ |
| $\alpha = 2$ | $0.95 \pm 0.15$ | $27.11 \pm 1.05$ |
| $\alpha = 5$ | $3.87 \pm 1.08$ | $232.58 \pm 1.78$ |
| $\alpha = 10$ | $5.72 \pm 1.19$ | $703.62 \pm 19.29$ |
| $\alpha = 20$ | $22.99 \pm 15.52$ | $2341.43 \pm 52.46$ |

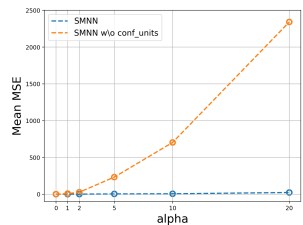

Figure 4: Visualization of the results in Table 11

### C.4 ADDITIONAL SCALABILITY TEST WITH RESPECT TO NUMBER OF MONOTONIC FEATURES

We conducted an additional scalability test with respect to the number of monotonic features, for the case where the number of monotonic features $m$ is dominantly larger than the number of non-monotonic features. Specifically, we created a 220-dimensional dataset, excluding the output feature, consisting of 20 uniform distribution-based dummy inputs and 200 monotonic inputs achieved by expanding each group of features of equation 11 from 5 to 50 features. Notably, in this experiment, the dummy features exert much less influence during training compared to our original experiment. The training time and MSE results are shown in Table 12 and Fig.5. We observed the consistent trend of training times remaining constant while MSE decreased as the number of monotonic inputs increased.

Table 12: Scalability test with respect to increasing number of monotonic feature when $m = 200$

| Network | # Parameters | Test MSE | Time (s) |
|---|---|---|---|
| SMNN ($m = 10$) | 29414 | $0.204 \pm 0.12$ | $78.76 \pm 2.22$ |
| SMNN ($m = 20$) | 29977 | $0.216 \pm 0.12$ | $78.40 \pm 2.57$ |
| SMNN ($m = 50$) | 31657 | $0.175 \pm 0.09$ | $78.76 \pm 1.51$ |
| SMNN ($m = 100$) | 34457 | $0.117 \pm 0.09$ | $79.08 \pm 2.06$ |
| SMNN ($m = 200$) | 40057 | $0.013 \pm 0.01$ | $78.84 \pm 2.01$ |

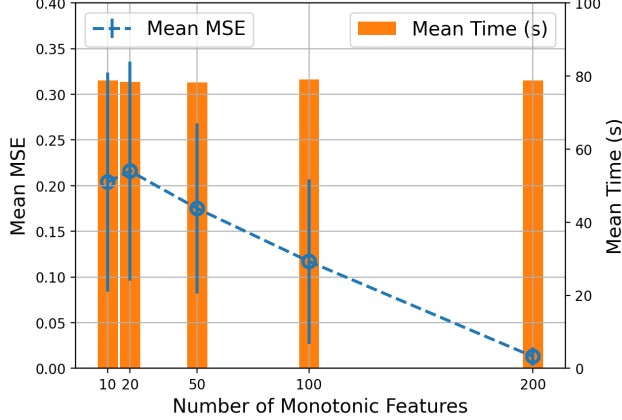

Figure 5: Visualization of the results in Table 12

