# OpenReview forum: "Scalable Monotonic Neural Networks"
_ICLR.cc/2024/Conference — ICLR 2024 poster_

### Official Review · Reviewer_y34F · 2023-10-24

**Soundness:** 3 good
**Presentation:** 3 good
**Contribution:** 3 good
**Rating:** 8
**Confidence:** 3

**Summary:**

Authors propose a new method, Scalable Monotonic Neural Networks (SMNN), to learn monotonic neural networks. Monotonic features are propagated forward using exponentiated units, which guarantees monotonicity. Non-monotonic features are propagated forward through confluence units and ReLU units, where confluence units capture interactions between monotonic features and non-monotonic features, and ReLU units capture interactions among non-monotonic features. The authors provide a theoretical result to verify the monotonicity of SMNN and experimental results to demonstrate the performance of SMNN.

**Strengths:**

1. The investigated problem, learning monotonic neural networks, is important in many fields, where some features are believed to have a positive or negative impact on the concerned output.
2. To the best of my knowledge, the proposed SMNN is novel.
3. The proposed SMNN is succinct. It is easy to understand as it only consists of commonly used activation functions, exponentiated weights, and a two-group architecture. Thus, SMNN can be trained by traditional backpropagation algorithms. It is also intuitive that SMNN can guarantee monotonicity.

**Weaknesses:**

1. The authors review previous methods in detail in the introduction part but the introduction might be too long and Table 1 does not fully distinguish SMNN and other methods. In Table 1, LMN, constrained MNN and SMNN have the same characteristics. It would be better to summarize the difference between SMNN and other methods in Table 1, which helps readers grasp the advantage of SMNN without reading through the long introduction.
2. The authors provide many experiments but it seems that the proposed SMNN has a similar performance to LMN. There are 5 datasets. SMNN wins on 2 of them, loses on 2 of them, and is comparable on the last one.
----
The authors' answers are convincing and I have updated my ratings.

**Questions:**

1. On page 2, the authors claim that the assurance of monotonicity is not always guaranteed as a regularization method. This is true but I think this should not be treated as the drawbacks of regularization methods. The monotonicity usually comes from experience and may not be the truth. When we prefer to believe the experience, we can use methods in the first group to guarantee monotonicity. But when the experience contradicts collected data, then regularization provides a chance to balance between past experience and new data. Thus, I think it is more suitable to say these two groups of methods have their applications in the real world rather than treating not guaranteed monotonicity as a drawback.
2. Authors provide two definitions for partial monotonicity, one using function values and one using partial derivatives. However, it seems that the definition using function values is not correct. I think the correct one should be "The function $f(\boldsymbol{x})$ is partially monotonically non-decreasing on $\boldsymbol{x}_m$ iff $\forall i, x_i \leqslant x_i' , \forall j \neq i , x_j = x_j' \Rightarrow f(\boldsymbol{x}) \leqslant f(\boldsymbol{x}')$. Take a 2-dimensional case as an example. Let the 1st coordinate be the monotonic feature, and the 2nd is not. Let $f(x_1 , x_2) = x_1 + x_2^2$. Then $f$ is partially monotonic according to the definition using partial derivatives but is not partially monotonic according to the definition using function values.
3. On page 4, the authors claim that "ReLU is commonly used as an activation function due to its advantages such as avoiding gradient vanishing". ReLU helps address the problem of gradient vanishing (compared with sigmoid) but it cannot "avoid" gradient vanishing. When the weight parameters are sufficiently small and the number of layers is large, ReLU also faces the problem of gradient vanishing, and other techniques are needed to help address the problem of gradient vanishing. Thus, it would be better to use the verb "alleviate" rather than "avoid".
4. In Eq. (5), $h_{con}$ should be $h_{conf}$.
5. In the current SMNN structure, both the exp unit and conf unit use the ReLU-n activation function for the purpose of universal approximation and aligning output magnitudes. But in the ReLU unit, ReLU is used rather than ReLU-n. I am wondering what will happen if all these three units use the ReLU-n activation function. I think a unified activation function makes the structure more succinct.
6. In theorem 1, $x$ belongs to $\boldsymbol{x}_m$. To my understanding, $\boldsymbol{x}_m$ is a vector rather than a set, which indicates that the belonging relation is not rigorously defined. I think $x$ is a coordinate of the vector $\boldsymbol{x}_m$. A suitable way is to define the subscript set of $\boldsymbol{x}_m$ as $M$, and then use $x_i$ with $i \in M$.

---

> ### Author Response · Authors · 2023-11-16
>
> W1: Certainly, we appreciate your constructive feedback. Table 1 has been updated in response to your suggestion, incorporating an additional column labeled `trainable using traditional gradient descent.’ Notably, our method allows for training with the conventional gradient descent algorithm, while both LMN and Constrained MNN require intentional modifications to weights during their training, making traditional gradient descent infeasible. Furthermore, we have introduced a classification of benchmark techniques into two major groups for clarity.
>
> W2: Certainly, we acknowledge that our method demonstrated performance on par with existing methods. While achieving lower performance could be considered a weakness, we would like to emphasize that comparable performance should not be perceived as a weakness, in our perspective. While striving for better performance is always desirable, please understand that attaining the best performance in the field of monotonic neural networks is not the sole objective of this research. Our method, as highlighted in Table 1, presents advantages over existing techniques, maintaining competitive performance levels. Given that our approach supports gradient descent learning without the need for additional modifications (such as weight normalization or weight absolutization) or the imposition of mathematical constraints such as the Lipschitz constraint, we see our method as a valuable addition to the field of monotonic neural networks because it will be suitable for expanding research into different complex network architectures for future works, such as neural additive models or convolution structure.
>
> Q1: We deeply appreciate your valuable feedback. As we totally agree with your comment, we have made the following modification.
> Before: However, this method has some drawbacks. It requires additional processes to enforce and verify monotonicity, and the assurance of monotonicity is not always guaranteed as a regularization method.
> After: However, it requires additional processes to enforce and verify monotonicity. As a regularization method, this method does not always ensure monotonicity. Therefore, in scenarios where the guarantee of monotonicity is essential, opting for the approaches with customized architecture may be preferable.
>
> Q2: Again, we deeply appreciate your valuable feedback. As we totally agree with your comment, we have made the modification on the definition using function values. Please refer to the revised manuscript.
>
> Q3: This feedback is very considerate. Thank you so much for very carefully reviewing our paper. The word 'avoiding’ has been replaced with the word 'alleviating.’
>
> Q4: We have modified the equation by following your comment. Thank you.
>
> Q5: Thank you for raising an interesting issue. Please refer to the newly created Appendix C1 on this matter. Additionally, we have inserted the following sentence into the middle of the 'Network structure’ paragraph.
> `` To maintain a consistent activation function throughout the network, it is acceptable to use the ReLU-n function for the ReLU unit (See Appendix C1.).’’
>
> Q6: We believe that your comments have significantly improved our paper. We have made modifications to Theorem 1 based on this feedback.

---

> > ### Comment · Reviewer_y34F · 2023-11-21
> >
> > Thanks for the detailed answering. The revised version and answers solve all my concerns. Especially, the revised Table 1 clearly demonstrates the difference between the proposed method and traditional ones, which also implies the potential advantage of the proposed method. After reading other reviewers' comments and authors' corresponding feedback, I find no obvious weakness. Thus, I have updated my ratings and suggest acceptance for this paper.

---

> > > ### Author Response · Authors · 2023-11-21
> > >
> > > We are pleased that you are satisfied with our responses. We are truly grateful for your dedicated time and effort spent on reviewing our paper.

---

> > > > ### Comment · Reviewer_y34F · 2023-11-23
> > > >
> > > > I have some further comments to share with and be confirmed by authors. It would be great if authros can provide their feedback on these ideas.
> > > > 1. ReLU-n is used in exp unit since ReLU cannot achieve universal approximation while ReLU-n can. I think another motivation is the gradient problem. If we use ReLU in exp unit and $exp(w)^\top x$ is large, then a small change of $exp(w)^\top x$ will cause a large change of the output, which may suffer from gradient exploding problem. Thus, it seems that the truncation is necessary to alleviate gradient exploding when using exp unit.
> > > > 2. While exp unit guarantees monotonicity, it is hard for exp unit to handle useless feature. If a feature $x$ has no impact on the output, then in traditional neuron, its weight usually converges to 0. But in exp unit, the weight needs to converge to $- \infty$, which may cause instability. Based on this, it is important to carefully choose monotonic features. This also provides some insights on the method mentioned by Reviewer sRQA: learning all features with exp unit may be difficult when there exist useless features.

---

> > > > > ### Author Response · Authors · 2023-11-23
> > > > >
> > > > > We sincerely appreciate you raising further discussion points.
> > > > >
> > > > > [1] Certainly, we opted for the ReLU-$n$ function not only for its capability in universal function approximation but also to mitigate the gradient exploding issue you highlighted.
> > > > >
> > > > > [2] Thank you for bringing up an interesting point regarding training with irrelevant input features. Your observation about the weight converging to negative infinity to make its exponentiated value zero is accurate and can indeed pose a significant challenge in neural network training. To address this concern, we outlined our weight initialization strategy in Appendix A3. However, as it is yet certain, we acknowledge that the potential impact of exponentiated weights on identifying irrelevant features is an interesting and important aspect that requires further investigation in the future.

---

> > > > > > ### Comment · Reviewer_y34F · 2023-11-23
> > > > > >
> > > > > > Thanks for your response. I have no further questions. : )

---

### Official Review · Reviewer_iEFi · 2023-11-01

**Soundness:** 3 good
**Presentation:** 3 good
**Contribution:** 2 fair
**Rating:** 5
**Confidence:** 5

**Summary:**

The paper tackles the inductive bias of monotonicity in Neural Networks. The authors propose a module based on nonnegative weights with a ReLU-n activation to enforce monotonicity. The proposed architecture comprises three types of hidden layers to handle monotonic and non-monotonic inputs.

**Strengths:**

- The paper is very clear and easy to follow.
- The method itself is simple and easy to implement.
- Variety of experiments across several domain-specific datasets and toy problems.

**Weaknesses:**

- It could be argued that the novelty of the approach is limited since Mikulincer & Reichman (2022) already describe a very similar architecture and prove the universality of 4-layer threshold networks with nonnegative weights in approximating monotonic datasets. ReLU-n is a continuous relaxation of the threshold function, so it is likely that universality holds here as well for the same reasons and might possibly offer better convergence properties, but this is not explored.
- There seems to be a big focus on scalability, but it’s unclear how this approach is better than current state-of-the-art models in this respect. The experiments do not compare against other models, and for good reason, I imagine; there is little difference in the scalability of the proposed method and existing works in the literature. So, this begs the question of why scalability is a central thesis in the paper.

Overall, this paper feels like yet another approach to the inductive bias of monotonicity in neural networks. It has no big flaws, but the advantages do not appear to be substantial compared to current methods. The approach is simple enough and is worth considering, but the paper is slightly below the acceptance threshold.

**Questions:**

- Could you clarify what this sentence means: “Nevertheless, it should be noted that an optimally trained network is not always guaranteed with respect to the Lipschitz constant λ, particularly for larger constants”?
- I am unsure why scalability is considered such a central point in the paper. Some of the other methods are clearly just as scalable, including LMNs and constrained monotonic networks. Am I missing something?

**********Nits:**********

- Missing slash for $\exp$ instead of $exp$ in the proof of theorem 1 on page 5.

---

> ### Author Response · Authors · 2023-11-16
>
> W1: We sincerely acknowledge your point. Indeed, as you mentioned, the research by Mikulincer & Reichman (2022) demonstrated the universal approximation property for 4-layer monotonic neural networks. Yes, this research served as a motivation for our approach. Due to the reasons outlined below, we believe there are distinct differences between the two studies and that our proposed method has its own contribution.
> The conclusion of Mikulincer & Reichman (2022) states the following. ``One aspect we did not consider here is learning neural networks with positive parameters using gradient descent. It would be interesting to examine the efficacy of gradient methods both empirically and theoretically. Such study could lead to further insight regarding methods that ensure that a neural network approximating a monotone function is indeed monotone. Finally, we did not deal with generalization properties of monotone networks: Devising tight generalization bounds for monotone networks is left for future study.’’ As evident from their conclusion, they did not provide empirical verification through actual implementation, nor did they validate learning using gradient descent methods. Remarkably, our SMNN method directly translates their theoretical findings into practical implementation. As such, we firmly believe that our work stands at the forefront of the latest research advancements. While they did not propose a specific network structure and did not consider partially monotonic cases, we addressed both aspects in our study.
>
> W2: We have performed the experiments for scalability comparison. Please refer to the next paragraph. We genuinely appreciate your insightful feedback. While it is a fact that our method exhibits scalability when compared to benchmark methods, the primary reason for our use of the term 'scalability’ is the absence of prior claims in this particular aspect within existing research. We believed that introducing this term could contribute to differentiating our work. Furthermore, as demonstrated, our method remains scalable with respect to both increasing numbers of monotonic features and expanding network sizes. We are encouraged by the agreement on this scalability aspect expressed by some reviewers. We wish your perspective on extending the term 'scalability’ in a broader context. Our method's generic nature, as highlighted by other reviewers, positions it well for future research endeavors exploring diverse and complex network architectures, such as neural additive models or convolution structures. Your thoughtful consideration of these aspects will be highly appreciated.
> Indeed, creating a fair experimental environment for scalability comparison, considering different learning ways, structures, and feasibility based on the number of monotonic inputs, poses challenges in ensuring identical network sizes for training/test times. Nevertheless, we made efforts to assess the speed of our method in comparison to benchmarking methods, and the results are presented in the table below. While this is not a comprehensive comparison, it is evident that our method was faster than others based on the available comparisons (Notice that verifying time is not test time but additionally required time only for Certified MNN. We did not perform the comparison of test times as it is unnecessary because all methods except for COMET require a similar small inference time.). We hope this outcome addresses your concern. As we have already demonstrated the scalability of our method empirically in Section 4.1, we have chosen not to include the table below in the revised manuscript.
> |Dataset|Method|# Parameter|# Monotonic Features|Training Time (s)|Verifying Time (s)|
> |:---:|:---:|:---:|:---:|:---:|:---:|
> |Auto-MPG|Certified MNN|$11006$|$3$|$99.88\pm13.37$|$28.83\pm12.89$|
> ||COMET|$421$|$3$|$--$|$--$|
> ||HLL|$14301$|$3$|$113.64\pm1.89$|$--$|
> ||LMN|$14025$|$3$|$22.20\pm1.31$|$--$|
> ||Constrained MNN|$14025$|$3$|$26.04\pm1.20$|$--$|
> ||SMNN (Ours)|$14193$|$3$|$\mathbf{17.00\pm0.91}$|$--$|
> |Heart Disease|Certified MNN|$7318$|$2$|$27.16\pm6.71$|$11.31\pm6.71$|
> ||COMET|$785$|$2$|$--$|$--$|
> ||HLL|$7788$|$2$|$16.56\pm0.92$|$--$|
> ||LMN|$7617$|$2$|$10.04\pm1.17$|$--$|
> ||Constrained MNN|$7617$|$2$|$12.04\pm0.17$|$--$|
> ||SMNN (Ours)|$7905$|$2$|$\mathbf{9.40\pm0.91}$|$--$|

---

> ### Author Response · Authors · 2023-11-16
>
> Q1: Intended meaning of the sentence is as follows. Due to the constraint imposed by $\lambda$ in LMN, when the Lipschitz constant $\lambda$ is smaller than the intrinsic data monotonicity, it is impossible to sufficiently learn the given data using LMN because the inherent monotonicity is greater than $\lambda$ (LMN cannot approximate any function with a gradient exceeding the given Lipschitz constant). As you mentioned, it seems that the original sentence does not accurately convey the intended meaning. Therefore, in order to clearly convey the meaning, we have revised the sentence as follows.
> ``Nevertheless, it is important to note that if the Lipschitz constant \lambda remains smaller than the inherent monotonicity scale within data, LMN may struggle to effectively learn from the data. This limitation arises because LMN cannot approximate any function that holds a gradient exceeding the given Lipschitz constant.’’
>
> Q2: Please refer to our answer to W2.
>
> N1: Thank you for the comment. We have made modifications to the exponential function by following your feedback.

---

> > ### Comment · Reviewer_iEFi · 2023-11-22
> >
> > Thank you for the thoughtful response. I appreciate the effort that went into the reply, yet my original conclusion still holds: while this paper has no immediate weaknesses, the contributions remain limited. The approach has little technical or theoretical novelty and does not represent a significant improvement over prior works in any particular area. The scalability argument alone is insufficient to set this approach apart, especially because some baselines are just as scalable. The table above seems rather inconclusive (e.g., LMN and SMNN are within error bars in dataset 2), though I want to highlight that I don't think scalability should be a point of contention in the first place. There is no reason LMNs and SMNNs should have different scaling. Additionally, the original experiments in the paper show virtually no performance gap between the two approaches.
> >
> > In short, this paper is a new implementation of monotonic networks and does what it's supposed to do. However, I think my original score remains appropriate because SMNN does not excel in any specific aspect.

---

> > > ### Author Response · Authors · 2023-11-22
> > >
> > > We appreciate your thoughtful evaluation of our research and the reasoning behind the assigned score. Your decision is respected, and we value your perspective. While we may not contest your viewpoint regarding the relative superiority of our SMNN method compared to the most recent state-of-the-art methods in terms of performance and training time (although we firmly believe in its speed and accuracy), we would like to underscore the distinctiveness brought about by the trainability of our method using conventional gradient descent. This feature positions our method as scalable, enabling extension to ongoing research, e.g. development of a neural additive model using the SMNN structure. This work is currently in progress, aiming to create a model that combines the advantages of a neural additive model, such as interpretability (currently challenging for LMN or Constrained MNN), with monotonicity. Regardless of the outcome of the acceptance/rejection at the ICLR conference, we remain committed to advancing our research series, including SMNN and its extensions to various models, through (hopefully) participation in this conference and others, as well as through journal publications. We deeply appreciate the time and effort you dedicated to reviewing our paper and providing insightful comments.

---

> > > > ### Comment · Reviewer_iEFi · 2023-11-22
> > > >
> > > > Thank you for the response. You mentioned using a neural additive model for interpretability. It would have been great to see a full study in this direction, and I think it would strengthen the submission substantially. However, I see no reason why one couldn't use LMN modules in a neural additive model. Could you elaborate on that?

---

> ### Author Response · Authors · 2023-11-22
>
> Firstly, we sincerely appreciate your positive feedback on the idea of a monotonic neural additive model (NAM) and your engagement in a further discussion. We would like to clarify that our thoughts shared here are preliminary and based on our current understanding at an idea level.
>
> Regarding our comment about it being ‘challenging for LMN,’ we want to clarify that we did not imply it is impossible. The challenge we mentioned to is associated with the Lipschitz constant ($\lambda$). In a basic NAM structure, each subnetwork necessitates its own $\lambda$ for its specific input feature, requiring $k$ hyperparameters for $k$ input features. While we believe there is an optimal combination of $k$ $\lambda$'s, discovering it poses a substantial challenge. Moreover, in a higher-order NAM structure or a multitask NAM structure, determining the optimal $\lambda$ values becomes even more difficult, as the number of subnetworks remarkably increases. This is the basis for our thoughts of it being challenging for LMN.

---

### Official Review · Reviewer_sRQA · 2023-11-02

**Soundness:** 3 good
**Presentation:** 3 good
**Contribution:** 3 good
**Rating:** 8
**Confidence:** 3

**Summary:**

The paper presents an architecture for a neural net that guarantees the monotonicity of the output wrt a subset of predefined inputs, and still allows interactions between features derived from all inputs.

**Strengths:**

Originality
--------------
The method is related to several existing pieces of work, but the design of the specific architecture, enabling the integration of information from the non-monotonous inputs in a principled way, at every level, is original to my knowledge.
The architecture is generic enough that it should be applicable in a wide variety of settings.

Quality
----------
Experiments are well designed and match well with the (implicit) research questions, demonstrate well the behaviours of the algorithm, in particular:
1. when scaling up the number of parameters
2. when scaling up the number of monotonic features
3. when increasing the noise (compared to less-constrained or regular MLPs)

Experiments on real datasets show that this method is at least competitive with state-of-the-art methods.

Clarity
---------
The paper is clearly written, does not pose any notable challenge for comprehension. The method is defined clearly enough to be reimplemented from the paper, even without the provided source code.

Significance
-----------------
A scalable, end-to-end learning method guaranteeing the monotonicity wrt some inputs would have an impact for ML application where interpretability, trust, or predictability are necessary.

**Weaknesses:**

Quality
----------
1. Regarding the theoretical aspect, and the proof of Theorem 1, an issue is that ReLU-n is not a differentiable function, contrary to the definitions in section 2. Its derivative is not defined at 0 or n. I think the conclusion is still correct due to it being continuous, and that both the left-derivative and the right-derivative at every point exist and are >=0, but the proof and definition should be updated.
2. The scaling of compute time wrt number of monotonic features cannot really be extrapolated from varying the dimensions from 1 to 20. It's likely that at this scale, the execution time is dominated by constant factors, and it's unclear how it would actually scale to "high-dimensional monotonic features", compared to Certified MNN or COMET for instance. Due to the architecture, it is likely that the scaling time _per training step_ would scale up as well as a regular MLP, however optimization issues could happen and slow down convergence, for instance. It would be more effective to demonstrate the scalability of this method by applying it on a problem too big for Certified MNN or COMET.
3. In Tables 2 and 3, statistical ties should be bolded, and be assessed by a statistical test taking into account the variance of _both_ distributions. Assuming the ± numbers indicate 95% CIs, there should be many more ties, and the conclusions about SMNN outperforming all other methods on regression tasks to not really hold. For instance, on AutoMPG, 7.44±1.20 and 7.58±1.20 should clearly overlap.
4. The COMPAS dataset is used despite ethical concerns, since these features were used by the original COMPAS system to produce a score that was unfair and biased. This work uses the dataset despite not explicitly aiming at producing unbiased decisions, or examining critically the trained system.

Clarity
---------
1. In Table 1, maybe indicate which methods are categorized as "regularization", and which ones are "customized architectures". Or, if "customized architecture" is synonymous with "end-to-end learning", maybe indicate it in the caption.
2. The circled "+" signs in Figure 1 is confusing if the operation is concatenation, not addition. Maybe it would be clearer to have both arrows point directly at the next "Exp unit"? (Idem for the fc layer)
3. Why are Table 2 and 3 separate? Both have a mix of regression and classification datasets, but they have a different set of methods, and those in Table 3 do not report parameter counts.
4. Not clear at first what "subjected to denoising" (S. 4.1) means.
5. Clarify what the "±" numbers in the table means: standard deviation, confidence interval?

Significance
-----------------
The small scale of experiments is a limitation. Given that the ambition of the algorithm is to be "scalable", I think it should try to demonstrate scaling where some other algorithms are limited.

Minor points
-----------------
1. Datasets should have citations or footnotes when they are introduced in the main text, not only in the appendix. Maybe also refer to the appendix explicitly for which features were considered monotonic.

**Questions:**

1. To disentangle the effects of the connectivity pattern from the ones of monotonous parametrization, I'd be curious to see the performance of a network composed only of Exponential units (all fully connected), but where the non-monotonous inputs would be duplicated as [x_¬m, -x_¬m]. Is that something you tried?

Update after discussion
--------------------------------
Many points have been addressed, so I'm moving my score from 6 to 8.

**Details Of Ethics Concerns:**

The COMPAS dataset is used despite ethical concerns, since these features were used by the original COMPAS system to produce a score that was unfair and biased. This work uses the dataset despite not explicitly aiming at producing unbiased decisions, or examining critically the trained system.

---

> ### Author Response · Authors · 2023-11-16
>
> W1 (Quality): Thank you very much for pointing this out. According to your comment, we have modified the definition and the proof in Section 3 by using the left and right-hand limits.
>
> W2 (Quality): We acknowledge your concern. In response, we conducted an additional experiment involving the expansion of the model in equation (11). Specifically, we created a 220-dimensional dataset, excluding the output feature, consisting of 20 uniform distribution-based dummy inputs and 200 monotonic inputs achieved by expanding each group of features from 5 to 50 features. Notably, in this experiment, the dummy features exert much less influence during training compared to our original experiment. The training time and MSE results are shown in the table below. We observed the consistent trend of training times remaining constant while MSE decreases as the number of monotonic inputs increases. Please consider that the intention behind the scalability test using equation (11) was to demonstrate not only the constancy of training time but also the improvement in MSE performance with the gradual increase in monotonic inputs. We believe that this additional experiment adequately addresses your concern. However, since the original experiment effectively conveyed our intended message and presenting the equation for the 220-dimensional dataset might be challenging in the paper, we have chosen not to replace the original experiment with this new one.
> |# Monotonic Features|# Parameter|MSE|Time (s)|
> |:---:|:---:|:---:|:---:|
> |$m=10$|$29414$|$0.204\pm0.12$|$78.76\pm2.22$|
> |$m=20$|$29977$|$0.216\pm0.12$|$78.40\pm2.57$|
> |$m=50$|$31657$|$0.175\pm0.09$|$78.24\pm1.51$|
> |$m=100$|$34457$|$0.117\pm0.09$|$79.08\pm2.06$|
> |$m=200$|$40057$|$0.013\pm0.01$|$78.84\pm2.01$|
>
> W3 (Quality):
> We appreciate the constructive feedback. To enhance the statistical clarity of our results, we have made revisions to Tables 2 and 3, accounting for statistical ties. The values following the `±’ symbol in these tables indicate the standard deviation based on 25 repetitions of our experiments. As the references do not explicitly provide details on experimental configurations such as the number of repetitions, we lack the necessary information to conduct statistical tests for benchmarking methods. Consequently, we have defined a statistical tie when, for both methods, one mean falls within the range (mean±std) of the other. Instances of statistical ties by this definition are marked with a $\dagger$ symbol.
>
> W4 (Quality): We acknowledge the ethical concerns surrounding the use of the COMPAS dataset (we already mentioned this in Appendix A2.), particularly given its history of being utilized in ways that produced biased and unfair outcomes. While our work does not explicitly focus on producing biased/unbiased decisions or critically examining the trained system from a fairness perspective, we used the COMPAS dataset as a means to test and demonstrate the technical capabilities of our model in a widely recognized context. Please also notice that this dataset has been used in several published papers for the same purpose with ours. Nevertheless, if it is still unable to avoid ethical concerns, we will definitely stop using the data and remove its results.
>
> W1 (Clarity): Thank you for the constructive comment. By following the comment, we have revised Table 1. In addition, we have incorporated an additional column (trainable using traditional gradient descent) to illustrate the distinction between our approach and the LMN and Constrained MNN methods. Unlike our method, both LMN and Constrained MNN necessitate deliberate adjustments to weights during their training processes, rendering the utilization of the traditional gradient descent algorithm infeasible.
>
> W2 (Clarity): Thank you for your comment. The `+’ in a circle symbol generally represents concatenation in the related literature. To enhance understanding, Fig. 1 has been modified. The meaning of the symbol has been noted in the bottom right corner of the figure. We would be grateful if you could confirm whether this modification has clearly resolved the issue.
>
> W3 (Clarity): As mentioned in the main text, Tables 2 and 3 include experimental results from the existing benchmark studies. Therefore, the tables have been separated for entry (not dividing the tables based on classification/regression). In Table 3, the original benchmark literature 'COMET’ [Sivaraman et al., 2020] was published without specifying the number of parameters. And the studies 'LMN’ [Nolte et al., 2022] and 'Constrained MNN’ [Runje & Shankaranarayana., 2023] also missed the number of parameters. Therefore, unfortunately, we could only include the performances of benchmark methods for comparison. Please refer to the benchmark literatures.
>
> W4 (Clarity): We have modified the sentence as follows. ``Fig.3(c) presents an interesting observation where the noise of the test dataset was eliminated.’’

---

> > ### Author Response · Authors · 2023-11-16
> >
> > W5 (Clarity): Please refer to our answer to W3(Quality).
> >
> > Sig.: For the proposed method only, please refer to our answer to W2(Quality). To address other reviewer’s concern, we have performed an additional experiment for training time comparison. Please refer to our answer to Q4 for the reviewer tddM. The limitations of the three techniques (Certified MNN, COMET, HLL) are apparent in terms of scalability, as observed in the comparative experiment. In the context of scalability, we find that the other three methods (LMN, Constrained MNN, SMNN) exhibit comparable characteristics. However, when considering scalability in the sense of extensibility, the proposed method stands out as sufficiently generic (unlike LMN and Constrained MNN), making it suitable for future research exploring various complex network architectures, such as neural additive models or convolution structures, as you pointed out.
> >
> > MP: Thank you for the comment. We have accordingly revised what you pointed out.
> >
> > Q: This is indeed an interesting and good idea, we think so. By following your idea, to address your curiosity, we have performed the experiments based on the Auto-MPG and Heart Disease datasets. The results below illustrate the comparability between your idea and our method.
> > SMNN : Auto-MPG(MSE) 7.44 ± 1.20, Heart disease(Test Acc) 0.88 ± 0.04
> > Reviewer's suggestion : Auto-MPG(MSE) 7.44 ± 1.57, Heart disease(Test Acc) 0.86 ± 0.04
> > (Now, our defense ;)) However, there seems to be a trade-off in terms of scalability and model interpretability, as non-monotonic features would need to be manipulated and duplicated. While your idea is innovative, it is necessary to conduct a further analysis of the impact of feature duplication during the learning process.

---

> ### Comment · Reviewer_y34F · 2023-11-21
>
> I think it is a great and potential idea to use only exp units but duplicated non-monotonic inputs. At least from the aspect of formulation, it provides a unified and simple way to guarantee monotonicity.

---

> ### Comment · Reviewer_sRQA · 2023-11-22
>
> Thanks for the comprehensive updates, they address most of the points raised.
>
> I think these are the 2 main remaining weaknesses:
> 1. Scalability is presented as one of the main advantages of this method compared to SOTA, but the paper does not demonstrate cases where, because of scalability, this method actually outperforms others (either because other methods become infeasible or too expensive, or because their performance drops).
> 2. The architecture introduces some complexity (confluence units, specific connectivity) and hyperparameters (relative size of the 3 ReLU / confluence / exp sub-layers at each layer), but the paper does not make a strong case against simpler variants.
>
> Minor points:
> 1. The updated proof of monotonicity is not complete, as it does not show that the resulting function `g` is continuous (nor does it use the fact that ReLU-n is). For instance, a function like $x \mapsto x - \lfloor x \rfloor$ would satisfy your proof, but would not actually be monotonic.
> 2. It would be nice to have the additional tables you pasted here in the Appendix, even if you chose not to replace the ones in the main paper.

---

> > ### Author Response · Authors · 2023-11-23
> >
> > We are glad you found our responses satisfactory and appreciate your additional discussion points.
> >
> > W1: As pointed by you and the other reviewers, it is acknowledged that our method might exhibit comparable training/test times with LMN and Constrained MNN, making it challenging to demonstrate a clear superiority. However, in response to the feedback from Reviewer iEFi, we emphasized that our method lends itself to seamless extension into more sophisticated research works, like integrating SMNN with a neural additive model or potentially other approaches. We encourage you to consider this as an expansion of the concept of scalability. For a more detailed discussion on this matter, please refer to the discussion thread with Reviewer iEFi.
> >
> > W2: Again, your suggestion for training a monotonic neural network is truly valuable. We will definitely consider the idea in our future research. However, relying solely on the exponential unit, while the simplest variant, might encounter limitations as mentioned recently by Reviewer y34F, particularly being difficult to make irrelevant weights zero. Additionally, the size of network remarkably increases as the number of non-monotonic features increases. Thus, we believe that the proposed SMNN method retains its merits in this context.
> >
> > M1: Certainly, your observation is accurate. Indeed, in presenting our ideas within the neural network context, the continuity of both the SMNN function $f(x)$ and the function $g(x)$ was implicitly assumed. However, in order to explicitly clarify monotonicity, as per your suggestion, we have revised the definitions for monotonicity and Theorem 1 to explicitly mention the continuity of the functions.
> >
> > M2: We appreciate your suggestion. The results for the additional experiment involving a larger number of monotonic inputs have been presented in Appendix C4.

---

### Official Review · Reviewer_tddM · 2023-11-02

**Soundness:** 2 fair
**Presentation:** 2 fair
**Contribution:** 2 fair
**Rating:** 5
**Confidence:** 3

**Summary:**

This work proposes Scalable Monotonic Neural Network (SMNN) to learn neural networks that preserve monotonicity of a model w.r.t. a subset of inputs. Existing works require additional overhead in terms of inference cost, weight constraints during training, scalability issues w.r.t. network size and number of monotonic inputs, etc. This work bakes in the monotonic properties directly into the network architecture by designing hidden layers with three different units ( exponential unit, ReLU unit and confluence unit ). Exponential unit preserves the monotonic nature of its input and are explicitly tied to the monotonic inputs. Remaining inputs go through the confluence and the ReLU unit, while the output of the confluence unit also goes through addition with the input for the next exponential unit (see Figure 1). Thus, monotonic inputs pass through a unique path in the network that preserves their monotonic nature. Empirical evaluation demonstrates that this method achieves comparable performance to the existing state-of-the-art and helps eliminate many issues arising in those works.

**Strengths:**

- Interaction of the monotonic inputs is segregated from the other inputs through a dedicated path in the neural network. This ensures the monotonic nature as these inputs pass through exponential units in the hidden layers
- Empirical experiments show the viability of such a simple approach to enforcing monotonicity

**Weaknesses:**

- Only performs on-par as the existing methods for enforcing monotonicity in neural networks
- Its unclear if there's any computational advantage (in terms of training/inference cost) compared to existing baselines
- Limited discuss on extensions to other activations or expressivity of the network

**Questions:**

- Does restricting interaction between monotonic and non-monotonic inputs hurt expressivity of this network?
- How does one incorporate other activations/non-linear operators in this architecture?
- How do you select the parameter n in ReLU-n activation?
- Do you have any comparison on train/test time for the baselines with the SMNN architecture? Does it take longer to converge compared to existing methods?
- Why are parameters missing for XGBoost and Isotonic methods in Table 2?

---

> ### Author Response · Authors · 2023-11-16
>
> W1: Certainly, we acknowledge that our method demonstrated performance on par with existing methods. While achieving lower performance could be considered a weakness, we would like to emphasize that comparable performance should not be perceived as a weakness, in our perspective. While striving for better performance is always desirable, please understand that attaining the best performance in the field of monotonic neural networks is not the sole objective of this research. Our method, as highlighted in Table 1, presents advantages over existing techniques, maintaining competitive performance levels. Given that our approach supports gradient descent learning without the need for additional modifications (such as weight normalization or weight absolutization) or the imposition of mathematical constraints such as the Lipschitz constraint, we see our method as a valuable addition to the field of monotonic neural networks because it will be suitable for expanding research into different complex network architectures for future works, such as neural additive models or convolution structure.
>
> W2: Please see our response to Q4.
>
> W3: Please see our responses to Q1, Q2, and Q3.
>
> Q1: Our answer to your question is yes. To investigate the effect of the confluence unit, specifically designed for handling interactions between monotonic and non-monotonic features, we conducted an ablation study and integrated its outcomes into Appendix C3 of the revised manuscript. The following sentence was accordingly modified in the `Network structure’ section of the revised manuscript.
> `` the confluence unit combines the outputs of the exponentiated and ReLU units to capture interactions between both types of features (See Appendix C3 for the justification of the confluence
> unit.).’’
>
> Q2: We appreciate the constructive feedback regarding the extension for the activation function. In order to address this concern, we added the following sentences at the end of the paragraph of `Activation functions’,
> `` Alternative activation functions offering the similar benefit, including both convexity and concavity, can be substituted for the ReLU-$n$ function (See our ablation study in Appendix C1.).’’
> and the ablation study in Appendix C1.
>
> Q3: Thank you for your valuable feedback. Indeed, the parameter $n$ in ReLU-$n$ is a hyperparameter to be predetermined as we mentioned in Appendix A1. We consistently set it to 1 throughout all our experiments, because the primary purpose with this activation function was to utilize the convex and concave properties, making detailed tuning of this value unnecessary. Nonetheless, to empirically investigate whether the performance of our method holds across different values of $n$, we conducted an ablation study and integrated its outcomes in Appendix C2 of the revised manuscript.
>
> Q4: Indeed, creating a fair experimental environment for comparison, considering different learning ways, structures, and feasibility based on the number of monotonic inputs, poses challenges in ensuring identical network sizes for training/test times. Nevertheless, we made efforts to assess the speed of our method in comparison to benchmarking methods, and the results are presented in the table below. While this is not a comprehensive comparison, it is evident that our method was faster than others based on the available comparisons (Notice that verifying time is not test time but additionally required time only for Certified MNN. We did not perform the comparison of test times as it is unnecessary because all methods except for COMET require a similar small inference time.). We hope this outcome addresses your concern. As we have already demonstrated the scalability of our method empirically in Section 4.1, we have chosen not to include the table below in the revised manuscript.
> |Dataset|Method|# Parameter|# Monotonic Features|Training Time (s)|Verifying Time (s)|
> |:---:|:---:|:---:|:---:|:---:|:---:|
> |Auto-MPG|Certified MNN|$11006$|$3$|$99.88\pm13.37$|$28.83\pm12.89$|
> ||COMET|$421$|$3$|$--$|$--$|
> ||HLL|$14301$|$3$|$113.64\pm1.89$|$--$|
> ||LMN|$14025$|$3$|$22.20\pm1.31$|$--$|
> ||Constrained MNN|$14025$|$3$|$26.04\pm1.20$|$--$|
> ||SMNN (Ours)|$14193$|$3$|$\mathbf{17.00\pm0.91}$|$--$|
> |Heart Disease|Certified MNN|$7318$|$2$|$27.16\pm6.71$|$11.31\pm6.71$|
> ||COMET|$785$|$2$|$--$|$--$|
> ||HLL|$7788$|$2$|$16.56\pm0.92$|$--$|
> ||LMN|$7617$|$2$|$10.04\pm1.17$|$--$|
> ||Constrained MNN|$7617$|$2$|$12.04\pm0.17$|$--$|
> ||SMNN (Ours)|$7905$|$2$|$\mathbf{9.40\pm0.91}$|$--$|
>
> Q5: As mentioned in the main text, the results in Table 2 came from the references (Liu et al., 2020 ; Nolte et al., 2022; Runje & Shankaranarayana, 2023). The paper where these experimental results for XGBoost and Isotonic were first presented is [Liu et al., 2020] and the number of parameters for XGBoost and Isotonic were missing in the original paper. This may be because they are not based on neural networks. We could not include the number of parameters for those methods.

---

### Author Response · Authors · 2023-11-22

We deeply appreciate the diligent efforts of all the reviewers and their valuable feedback on our paper. We have provided comprehensive responses to the concerns raised by all the reviewers, and we are grateful for the thoughtful opinions shared by two of them.

However, we note that the other two reviewers have yet to check our responses and revisions. Given the limited time remaining in the discussion period, which is less than a day, we kindly request the participation of these two reviewers in this discussion. Their engagement will allow us to ascertain whether our rebuttals effectively address their concerns. We understand the time constraints and sincerely thank everyone for their dedicated time and efforts in reviewing our paper.

Best regards, Authors

---

### Comment · Area_Chair_daHb · 2023-11-22

Dear all,

The author-reviewer discussion period is about to end.

@authors: If not done already, please respond to the comments or questions reviewers may further have. Remain short and to the point.

@reviewers: Please read the author's responses and ask any further questions you may have. To facilitate the decision by the end of the process, please also acknowledge that you have read the responses and indicate whether you want to update your evaluation.

You can update your evaluation positively (if you are satisfied with the responses) or negatively (if you are not satisfied with the responses or share other reviewers' concerns). Please note that major changes are a reason for rejection.

You can also keep your evaluation unchanged. In this case, please indicate that you have read the responses, that you do not have any further comments and that you keep your evaluation unchanged.

Best regards,
The AC

---

### Meta-Review · Area_Chair_daHb · 2023-12-09

**Metareview:**

The reviewers have mixed opinions about the paper, leaning either towards rejection (5-5) or strongly towards acceptance (8-8). The paper introduces a novel architecture for monotonic neural networks. The paper is well presented and the empirical evaluation is good. The results, however, do not reveal a strong advantage in terms of accuracy -- it performs as well as LMN, but not better. The author-reviewer discussion has been constructive. The authors argued in favour of acceptance by pointing out the better scalability of their approach. Training times are indeed 5-20% shorter. Overall, the opinions of the reviewers have remained mostly unchanged.

My recommendation is to accept the paper. Although the results are not significantly better than the SOTA, the paper contributes a novel architecture with interesting properties, which may be useful for other applications. The experimental validation is fair and correct. The authors are encouraged to include the additional experiments performed during the author-reviewer discussion period in the final version of the paper.

**Justification For Why Not Higher Score:**

Technically correct, but not a strong contribution.

**Justification For Why Not Lower Score:**

The contribution of the architecture is interesting in itself and could lead to promising follow-ups.

---

### Decision · Program_Chairs · 2024-01-16

Accept (poster)